# An Overview of Flood Risk Analysis Methods

**Daniel Constantin Diaconu** [1,2] [iD], **Romulus Costache** [3,4] [iD] **and Mihnea Cristian Popa** [2,5,*]

1. Department of Meteorology and Hydrology, Faculty of Geography, University of Bucharest, Bd. Nicolae Bălcescu No 1, 1st District, 010041 Bucharest, Romania; daniel.diaconu@unibuc.ro
2. Centre for Integrated Analysis and Territorial Management, University of Bucharest, 010041 Bucharest, Romania
3. National Institute of Hydrology and Water Management, București-Ploiești Road, 97E, 1st District, 013686 Bucharest, Romania; romulus.costache@icub.unibuc.ro
4. Research Institute of the University of Bucharest, 90-92 Sos. Panduri, 5th District, 050663 Bucharest, Romania
5. Simion Mehedinți–Nature and Sustainable Development Doctoral School, University of Bucharest, 010041 Bucharest, Romania
* Correspondence: mihnea.cristian.popa@drd.unibuc.ro

**Abstract:** Scientific papers present a wide range of methods of flood analysis and forecasting. Floods are a phenomenon with significant socio-economic implications, for which many researchers try to identify the most appropriate methodologies to analyze their temporal and spatial development. This research aims to create an overview of flood analysis and forecasting methods. The study is based on the need to select and group papers into well-defined methodological categories. The article provides an overview of recent developments in the analysis of flood methodologies and shows current research directions based on this overview. The study was performed taking into account the information included in the Web of Science Core Collection, which brought together 1326 articles. The research concludes with a discussion on the relevance, ease of application, and usefulness of the methodologies.

**Keywords:** flood; machine learning; geospatial techniques; Web of Science Core Collection; hydraulic modeling





## 1. Introduction

Floods represent a natural phenomenon that annually generates significant economic and human losses [1–3]. Researchers' constant concern is the risk of floods to which the population is exposed. Various scientific methods can determine the magnitude and impact of floods. The large variety of these methods provided, the wide range of primary data used, the computer methods of processing, and spatial representation lead to identifying an optimal approach or methodologies for the researched area's specific conditions.

Hydraulic models (from 1D to 3D) are the most used method worldwide to generate flood maps for different return periods. The values of maximum flow and water layer thickness are the primary input information for any of these models. In this regard, several high-performance widely used models have been developed: the Hydrologic Engineering Center's (HEC) River Analysis System model from the US Army Corps of Engineer HEC; the MIKE 11 hydraulic model, developed at the Danish Hydraulic Institute; or the MIKE 21, modeling the FLO 2D model for 2D models. In this regard, the following works can be mentioned [4–9].

A very important increase in interest was represented by the number of research papers that approached the flood susceptibility estimation using bivariate statistics and machine learning models. This domain is in continuous development and includes scientific works from [10–15].

However, some risk assessment research assessing the potential damage caused by floods produces flood maps combined with social and economic spatial data. This economic

assessment of losses involves many parameters, some very subjective, that interact in a complex and non-linear way [16–19].

A current trend in research and hydrological models includes real-time flood economic losses. The estimates are based on geospatial data to delineate the areas and targets affected by floods [18,20–22]. This article aims to present current flood analysis methods, highlighting the methodologies used, the research areas, the progress made in this field, and the accessibility of information. Highlighting the current research directions and the most important methods helps to select an optimal research method quickly and can also help in pointing out the major gaps that should be addressed in future studies.

## 2. Materials and Methods

The analysis is focused exclusively on the works that researchers have published in scientific articles, using the Web of Science Core Collection database until 1 October 2020. Articles containing TITLE-ABS-KEY (flood *) were searched, and finally, 28,819 research works that addressed various aspects of floods were identified. Further, a "Refine Results" filter was applied with the term (methodology *), and only 1694 papers remained for the future analysis. An additional analysis of the papers showed us that 368 articles did not meet the research requirements, because they were included in other research fields (biology, chemistry, computer science). Thus, the number of analyzed articles was reduced to 1326.

A scan of the resulting titles indicated that they generally corresponded to the proposed objective of analyzing the types of methodologies used in flood research. They are analyzed in the next stage, for which the selected articles were reviewed individually to keep only those that referred to the goal of our study (for example, topics such as the effects of floods on flora and fauna were eliminated).The articles were classified based on the main topics presented in their abstracts. They were then revised and combined with articles on similar topics. This was an iterative process, with abstracts revised several times to ensure that the articles were adequately selected according to the proposed objective criteria.

The analysis presents the papers identified based on our search criteria. The Web of Science Core Collection database was chosen for the quality of the publications and their impact in the development of research, regardless of the research field. The analysis showed that most researchers prefer English, but some papers were written in Spanish and French. Still, the number of publications written in these two languages was rather low. Thus, our search may have omitted papers that could contribute to the methodological development of flood assessment because they were not explicitly identified within our selection method.

## 3. Results

The selected papers were published between 1979 and 2020 (Figure 1). It is worth noting the increase in the number of publications after the year 2000. In the last decade, researchers managed to publish up to 201 papers per year, with a mean of 115 papers/year, and with the lowest number of 45 papers in 2010.

Due to their important negative effects, floods and their impact on society represent a priority in the work of researchers today. The Web of Sciences database revealed that over 1326 research papers mainly focused on this topic (Figure 1). Analyzing the distribution of papers within the Web of Science database, around 52.18% were located in the field of water resources, 26.97% in the multidisciplinary geosciences, 22.66% in the environmental sciences, 17.82% in civil engineering, 14.93 % in meteorology and the atmospheric sciences, 7.91% in environmental studies, and 4.3% in environmental engineering and remote sensing. The top ten domains ended with 3.71% in physical geography and 3.54% in geography.

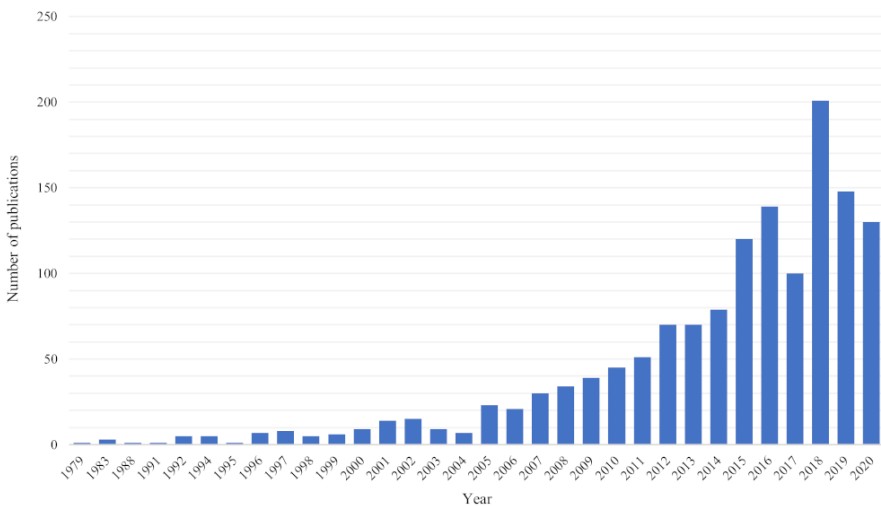

**Figure 1.** Number of publications per year.

Various institutions funded over 53% of the research. The institution with the largest number of published articles is currently the National Natural Science Foundation of China (NSFC), with 78 articles analyzed in our study, representing 4.60%. The Engineering Physical Sciences Research Council (EPSRC), the European Union (EU), and the Natural Environment Research Council (NERC) were identified with 44 published papers (2.59%). The National Science Foundation (NSF) had 30 articles (1.77%), the Natural Sciences and Engineering Research Council of Canada had 23 articles (1.35%), and the Spanish Government had 19 articles (1.12%). The European Commission Joint Research Center and the Federal Ministry of Education Research (BMBF) had 15 articles each (0.88%), closely followed by the Coordination for the Improvement of Higher Education Personnel (CAPES) and the Ministry of Education, Culture, Sports, Science, and Technology (MEXT) of Japan with 14 articles (0.82%). The China Scholarship Council, the Fundamental Research Funds for the Central Universities, and the Portuguese Foundation for Science and Technology followed with 13 articles (0.76%).

With a number between 11 (0.64%) and 7 articles (0.41%), we identified the National Council for Scientific and Technological Development (CNPQ), National Key Research and Development Program of China, Ministry Of Science and Technology Taiwan, National Key R&D Program of China, Spanish Ministry of Economy and Competitiveness, French National Research Agency (ANR), Ministry of Education, University and Research (MIUR) Research, National Basic Research Program of China, Australian Government, Interinstitutional Council Of Science and Technology (CICYT ), and Department for Environment Food Rural Affairs (DEFRA). The Chinese organizations had the highest number of articles on this topic, followed by Europe and the United States.

The most prolific authors were as follows: Bates, Paul with 21 articles (1.23%); Beven, John Keith with 17 articles (0.99%); Karamouz, Mohammad with 16 articles (0.94%); Hall, Jim W. with 13 articles (0.76%); Djordjević, Slobodan with 12 items (0.708%); Di Baldassarre, Giuliano, Ouarda, Taha, and Pappenberger, Florian with 11 items (0.64%), Borga Marco and Simonovic Slobodan P. with 10 items (0.59%), Bodoque, Jose Maria with 9 papers (0.53%), Herrero, Andrés Díez, Gouldby, Ben, Mediero, Luis, and Ping-an Zhong with 8 papers (0.47%), and Benito, Gerardo, Bobée, B., Chen, AS., Dottori, Francesco, Hong Yang, Jongman, Brenden, and Leandro, Jorge with 7 articles (0.41%) each.

The top journals that included the most published papers indicated both an interest in the topic and the prestige of the publishers and reviewers concerned with increasing the quality of published articles. On 1 October 2020, 104 articles (6.13%) relevant to our research were published. The following journals had the most published papers: *Natural Hazards* had 94 papers (5.54%), *Natural Hazards and Earth System* Sciences 66 articles (3.89%), *Journal of Flood Risk Management* 65 articles (3.83%), *Water* 61 articles (3.60%), *Water Resources*

*Research* had 50 articles (2.9%), *Disaster Prevention and Management* 38 articles (2.24%), *Hydrology and Earth System Sciences* 32 articles (1.889%), *Hydrological Processes* 30 articles (1.77%), *Science of the Total Environment* 28 articles (1.65%), *Journal of Hydrologic Engineering* 27 papers (1.59%), *Hydrological Sciences Journal (Journal Des Sciences Hydrologiques)* and *Water Resources Management* 25 articles each (1.476%), *International Journal of Disaster Resilience in the Built Environment and Sustainability* 21 articles (1.24%), *Remote Sensing* 20 articles (1.18%), *Houille Blanche Revue Internationale de L'Eau* and *Stochastic Environmental Research and Risk Assessment* 18 articles each (1.06%), *Environmental Earth Sciences* and *International Journal of Disaster Risk Reduction* 14 articles each (0.82%), *Advances in Water Resources* and *Water Science and Technology* 13 articles each (0.76%), and, last but not least, *Proceedings Of The Institution Of Civil Engineers Water Management* 12 papers (0.70%) and *International Journal Of Climate Change Strategies And Management* and *Journal Of Coastal Research* 11 articles each (0.64%).

Generally, the authors were affiliated with university centers, but there were also situations when they were part of institutes or private hydrological research companies. The following affiliations were prominent: the University of Bristol had a total of 35 articles (2.06%); the Delft University of Technology had 32 articles (1.88%); the University of Exeter had 25 articles (1.47%); Lancaster University had 25 articles (1.47%); the University of Tehran had 23 articles (1.35%); Consiglio Nazionale delle Ricerche (CNR) had 17 articles (1%); the European Commission, Newcastle University, Delft Institute for Water Education (formerly known as UNESCO IHE), the University of Oxford, and the University of Quebec had 15 articles each (0.88%); Middlesex University had 14 articles (0.82%); Deltares, Hohai University, the University of Coimbra, and Wuhan University had 13 articles each (0.76%); the Center for Ecology and Hydrology, Hydraulics Research (HR) Wallingford, the Indian Institute of Technology, the Polytechnic University of Milan, the University of Basilicata, the University of Bologna, the University of Bucharest, the University of Genoa, and the University of Oklahoma had 12 articles each (0.70%).

### 3.1. Spatial Distribution of Research

The research was mainly focused on the following three major regions: Continental Europe [22–26] with about 40% of articles, Asia with 28% [27–31], and North America with 20% [32–36]. South America (5%) [37–41] and Africa (4%) [42–46] showed less research interest (Figure 2). Following a detailed analysis of each region, we found that the most researched areas were: the United States, representing 13.19% of the total articles analyzed and 71.72% of the articles located in North America [47–49]; the United Kingdom, with 90 articles representing 17.92% of the articles in Europe and 6.78% of the total papers [50–52]; Italy and Spain, accounting for 5.27% of the entire articles and 13.94% of the scientific papers published in Europe, respectively [53–55]; and China, accounting for 5.27% of the total articles and 20.11% of the articles in Asia [56–58]. A significant percentage of publications also came from India, representing 16.6% of all Asian scientific articles and 4.37% of the total publications [59–61]. The latest analysis of publications showed us that the papers that have China and India as their study area are increasingly numerous.

The results of the analysis show the interest of researchers to test and validate the methodologies proposed with different research areas (West Africa, Balkans, South and Southeast Asia) [62–64]. Research was also conducted on a global scale, representing 1.05% [65–67].

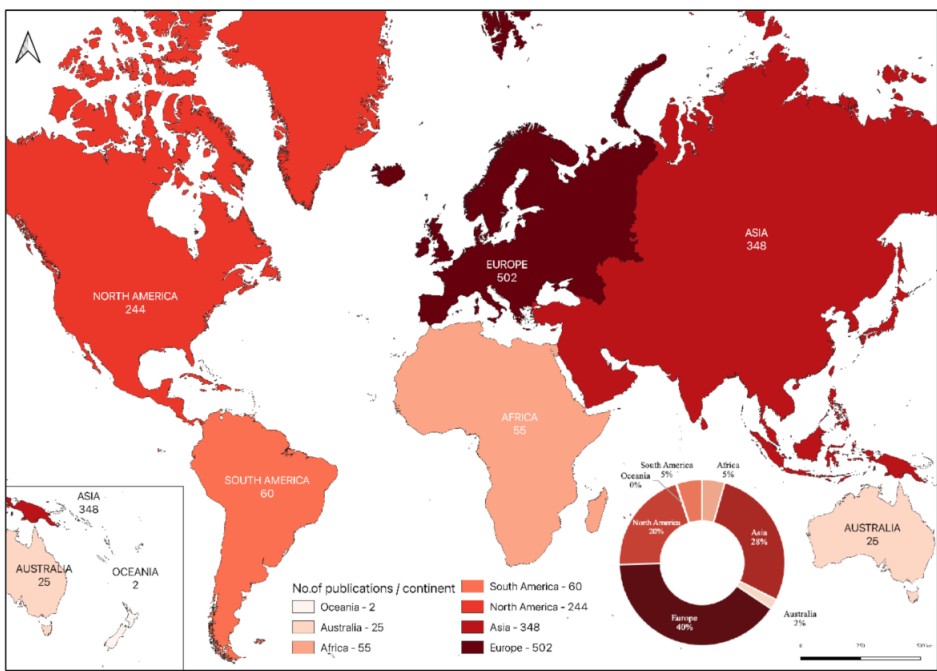

**Figure 2.** Spatial distribution of research with the number of publications per continent.

### 3.2. Research Methods and Advances in Flood Research

It is worth highlighting the variety of methods and their use in research. The detailed analysis of the papers allowed us to identify and classify the methodologies proposed within ten categories: remote sensing techniques, geographic information systems (GIS) techniques, combined methods (GIS—machine learning, GIS—modeling—simulation), modeling and simulation, statistical analysis, machine learning methods, comparison, survey and interviews, evaluation methods, and statistical and mathematical methods (Figure 3). Among these methodologies, the most popular methodological approach was that of modeling and simulation (hydrological, hydraulic, hydrodynamic), representing 24% of the total articles [68–73], followed by statistical analysis at 20% [74–78] and GIS techniques at 17% [79–84], while 16% of the articles proposed the use of combined methodologies [85–89]. The papers based on remote sensing techniques (analyzed via GIS environments) represented 8% of the total [90–94] and used ERS-1, Kompsat-2, LANDSAT, MODIS, and Sentinel-II imagery. Evaluation-based methodologies, consisting of framework proposals, reviews, and pre- and post-flood assessments, which have a theoretical aspect, represented 6% of the total articles [95–100].

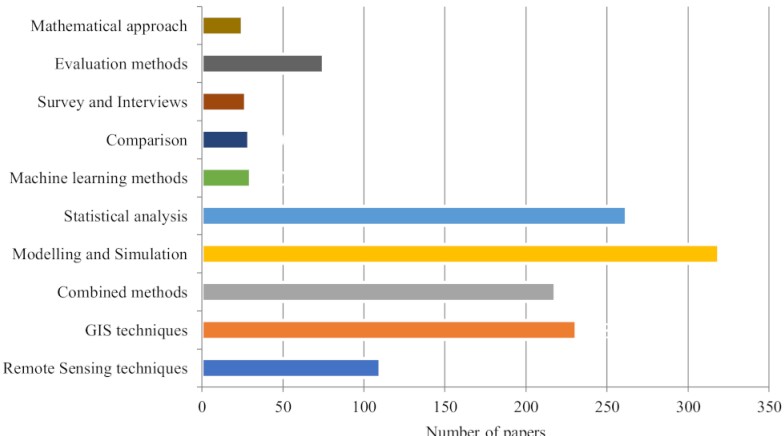

**Figure 3.** Number of papers per research category.

The role of geospatial technologies—remote sensing and GIS—has been a significant step in flood modeling, forecasting, and hazard assessment. Flood modeling requires the analyst to acquire, maintain, and widely use a spatial database. Remote sensing and GIS are excellent techniques that can meet these requirements.

The role of remote sensing in river basin hydrologic modeling as part of the entire runoff modeling process is due to its ability to provide continuous spatial data, measurements of hydrological variables that are not available through traditional techniques, and especially its ability to provide global data, e.g., long-term.

Modeling and simulation techniques refer to flood modeling as the process of transforming precipitation into a flood hydrograph. In this way, floods are approximated physically or mathematically (using mathematical equations) where the relationships between them represent the state at the system's entry and exit. Geographic information systems (GIS) have increased the importance of remote sensing by improving spatial modeling efficiency. This process has increased the ability to estimate hydrological models.

The hydrological model is a mathematical representation of hydrological processes in a river basin in a simplified form. It was used primarily to understand and explain hydrological processes and for hydrological prediction. This model can be of different types, depending on the kind of approach: deterministic or stochastic. A hydrological model is deterministic if it represents the physical processes in a river basin without considering randomness. It becomes a stochastic model when it incorporates in its random mathematical representation variables and their probability of distribution in the space of parameters.

On the other hand, the hydraulic models used show water flows' evolution to simulate water movement during floods, along waterways, reservoirs, and hydraulic structures. One of the most used hydrological and hydraulic modeling systems is the HEC river analysis system (RAS). The Hydrologic Engineering Center – Hydrologic Modelling System (HEC-HMS) is a generalized modeling system designed to simulate river basin dendritic systems' precipitation–runoff processes with a wide range of applicability.

The HEC RAS is an integrated software system designed to operate one-dimensional hydraulic calculations for a complete network of natural and artificial channels.

The combined methodologies use a mix of remote sensing and GIS techniques with those of modeling and simulation, statistics, or integration of machine learning algorithms. Recent years have shown us a diversification in the methodological approaches proposed by researchers, so that more and more research using methodologies based on machine learning (2% of articles) [100–103] and comparative methods (2% of total papers) [16,17,104–106] is starting to be published. The use of combined methods is also gaining more and more interest from researchers.

Mathematical theories were the first approaches to flood patterns, mainly using hydrological data. Based on existing data, new models have been created for areas with similar characteristics [107–109]. Statistics were used as soon as the databases allowed their processing with good results [110–114]. The development of GIS techniques has allowed the spatialization of these methodologies, leading to combined methods.

Statistical analysis was used mainly in estimating flood frequency. Until 1966, the most widely used distribution for flood frequency analysis was the type I extreme value distribution, also called the Gumbel distribution, followed by the lognormal. Both of the above distributions are still commonly used, commonly referred to in hydrology as Pearson type III and log-Pearson type III.

Tiago de Oliveira (1982) developed a statistical decision procedure to choose between the Gumbel, Frechet, and Weibull models. The fact is that hydrological data rarely fulfill the classical asymptotic theory's assumptions, and thus, the Gumbel, Frechet, or Weibull model is used.

Remote sensing techniques represent the beginning in this sense, so that later, the methods are based on GIS techniques, modeling, and simulation, and recently on machine learning methods [115]. The development of data processing techniques in a large volume

and the use of a large number of parameters has allowed for the development of new methodological approaches of high precision to mitigate the effects caused by floods. The spatialization of these methodologies using GIS techniques has opened the way for researchers to develop or improve existing methods but also to combine algorithms, thus leading to the creation of hybrid algorithms [17,105,106].

## 4. Discussion

The analyzed papers highlight the interest regarding the study of floods and methodologies for their most accurate analysis. Easy access to databases (e.g., European Environment Agency, United States Geological Survey (USGS), NASA, European Space Agency, Japanese Aerospace Exploration Agency (JAXA)) and high-speed information processing have led to the emergence of numerous publications in this field. At the beginning, survey and interview analysis methods and statistical–mathematical methods were used to study the characteristic parameters of floods; later on, the development of GIS, remote sensing, machine learning, modeling and simulation techniques contributed to the creation of new methods capable of providing a simulation of the phenomenon in different scenarios [106,116–122]. The technological progress has led to the development of combined methodologies and comparative approaches to highlight the effectiveness of the methods used and their ease of implementation in risk management systems and water resources. This finding also indicates the shortcomings of existing systems that cannot integrate all the parameters in the study of floods. There are also limitations generated by the infrastructure for obtaining data, transmitting, processing, and interpreting them. The development of these analysis methods also led to the formation of new analytical skills of specialists who generated new analysis methods, such as machine learning and comparison. At this time, the need to know the extent of the flood phenomenon to reduce the risk generated became evident. These methods, which are based mainly on the processing of geospatial information without requiring a high level of experience of those who use it, create disparities in the analysis of the phenomenon with the risk of results and methods inconsistent with the field reality [123–128]. Numerous methodologies that are presented as a novelty are not found in the current practical activity, as there is no validation of them in real-case scenarios. Some national organizations do not have the technical and economic support to implement such methods in their current activity [129–135]. Research into the effectiveness of the methods presented in the published articles and their implementation in the current applications of the profile organizations should also be encouraged by giving the minimum requirements of IT infrastructure (hardware and software), professional training, and access to specialized databases. New methodologies generally involve the aggregation of complex databases and high-resolution imagery, all generating the need for high-performance IT equipment but also for staff training.

The paper presents an overview of current research methodologies and identifies the methodological spectrum used in flood analysis, providing researchers with a general overview.

It should be noted, however, that the study has some limitations. There may be relevant additional research published in databases other than the one under consideration (Web of Science Core Collection), such as SCOPUS, African Journals Online, and Google Scholar.

## 5. Conclusions

The present study tried to highlight various aspects, methods, and approaches of flood study methodologies between 1979 and 2020.We analyzed 1326 articles of the most prolific researchers to make a solid and coherent analysis on the various methods used. The progress in the methodological framework and methodological approaches to floods was examined, highlighting the models used.

The research presents an overview of the methodologies used in global flood analysis. This wide range of methods used has advantages and disadvantages. Based on the expe-

rience of researchers specializing in the development of flood evolution methodologies, examples of good or bad practices in their current use can be developed later. It is difficult to compare the results presented by the numerous publications available to determine a generally standardized approach. Models set on determining databases have a higher accuracy, remote sensing techniques can advance outcomes over large areas of territory that are difficult to access, and machine learning methods offer advantages in data processing, the validation of results, or the integration of a large number of parameters. Therefore, the standardization of methodologies needs to be done at a regional level and depending on the scenario. Advances in geospatial techniques, especially in remote sensing, GIS, 3D hydrological models, and machine learning algorithms, have revolutionized the methodology of flood analysis.

Multidimensional approaches, models based on machine learning, high-resolution satellite imagery, hydraulic modeling, and a selection of efficient flood conditioning parameters are suggested for an in-depth hazard analysis. The use of multidimensional approaches, sophisticated models, site-specific indicators, and high-resolution satellite data are areas where further research is required.

**Author Contributions:** Conceptualization, D.C.D., R.C. and M.C.P.; methodology, D.C.D.; software, M.C.P.; validation, R.C.; formal analysis, D.C.D.; writing—original draft preparation, D.C.D. and M.C.P.; writing—review and editing, D.C.D. and R.C. All authors have read and agreed to the published version of the manuscript.

**Funding:** This research received no funding.

**Acknowledgments:** The authors gratefully acknowledge the anonymous reviewers for their helpful comments and suggestions to improve the previous version of the manuscript.

**Conflicts of Interest:** The authors declare no conflict of interest.

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
