# Peer review of "An Overview of Flood Risk Analysis Methods"

_water, doi:10.3390/w13040474_

Round 1

Reviewer 1 Report

The manuscript is interesting. Nevertheless, it needs some further improvements. In general, there are still some occasional grammar errors throughout the manuscript, especially the article "the," "a," and "an" is missing in many places; please make a spellchecking in addition to these minor issues. The reviewer has listed some specific comments that might help the authors further enhance the manuscript's quality.

  1. Specific Comments

Please include a list of acronyms and abbreviations.

Please provide more citable information in the abstract.

  • Introduction
  • The objectives should be more explicitly stated.
  • The authors need to enrich the background further. The following literature might be useful in this regard << Flood susceptibility modelling using advanced ensemble machine learning models>>, << Human–Environment Natural Disasters Interconnection in China: A Review>> and << Seasonality shift and streamflow flow variability trends in central India>> you may review other additional relevant references as well.
  • What is the novelty of this work?

  • Methods
  • The methodology limitation should be mentioned, such as literature exclusion, language limitation, databases, etc.

  • Results
  • This section needs to be entirely rewritten. Do not just summarize how many papers each institution has published that is not relevant; please provide more useful information.

  • Discussion
  • This section is quite poorly written. Please improve it. Compare your findings with other reviews, why your work stands out, what is the new contribution etc.
  • Also, the future research recommendation is missing.

Author Response

The authors thank you for your helpful comments and suggestions to improve the manuscript's previous version.

Reviewer 2 Report

Dear Authors,

your review could be an interesting piece of research, but the present version has too many flaws, that prevent the readers from catching what you really mean. In the attached pdf I reported some comments, but please restructure the whole research before resubmission.

Abstract: it is too vague. I suggest adding a few more details to attract readers.

Introduction: very general. Please provide a better review of the state-of-the-art, and point out the novelty of your approach, or why do you think that this study is needed.

Materials and Methods: as the previous sections, too vague. The methodology is not well described, and details are needed to assess the validity of the approach.

Results: is really helpful knowing which are the most cited sources of funds, authors and journals? Why? The quality of the results is highly affected by the lack of a detailed description of the used methods. I cannot judge the results as long as I do not know how (and why) you selected such articles.

Discussion: many claims reported in this section are not supported by the presented data. Please improve this section, addressing more in detail what you have reported (e.g., the geographical distribution of authors/studies, the funding sources, etc.) and the used methodology.

Conclusions: the conclusions are not supported by the rest of the paper. Please, rewrite the whole section.

I also suggest involving a native English speaker for double-checking the language.

Author Response

The authors thank you for your helpful comments and suggestions to improve the manuscript's previous version.

Dear Authors,

your review could be an interesting piece of research, but the present version has too many flaws, that prevent the readers from catching what you really mean. In the attached pdf I reported some comments, but please restructure the whole research before resubmission.

Abstract: it is too vague. I suggest adding a few more details to attract readers.

Thank you for your comment. We revised the abstract.

Introduction: very general. Please provide a better review of the state-of-the-art, and point out the novelty of your approach, or why do you think that this study is needed.

We added the following paragraphs in the introduction:

Between lines 44 and 51:

In this regard can be mentioned the works done by: Yang et al. (2006), Thakur et al. (2017), Thompson et al. (2017), Khalfallah and Saidi (2018), Dhote et al. (2019), and Huţanu et al. (2020).

A very important increase was represented by the number of research papers that approach the flood susceptibility estimation using bivariate statistics and machine learning models. This domain is in a continuous development and includes scientific works like: Chapi et al. (2017), Shafizadeh-Moghadam (2018), Termeh et al. (2018), Choubin et al. (2019), Costache et al. (2019), and Costache and Bui (2020).

Between lines 61 and 63:

Highlighting current research directions and the most important methods helps to select an optimal research method quickly, and can also help in pointing out the major gaps that should be addressed in the future studies.

Materials and Methods: as the previous sections, too vague. The methodology is not well described, and details are needed to assess the validity of the approach.

We have improved the Materials and Methods section by adding / modifying the following paragraphs:

Lines 70 - 79:

An additional analysis of the papers showed us that 368 articles did not meet the research requirements, these being included in other research fields (biology, chemistry, computer science). Thus, the number of analyzed articles was reduced to 1326.

A scan of the resulting titles indicated that they generally correspond to the proposed objective of analyzing the types of methodologies used in flood research, so they will be analyzed in the next stage, in which the selected articles were reviewed indi-vidually to keep those that refer to the goal of our study (for example, topics such as the effects of floods on flora and fauna, etc. were eliminated).

Lines 84 – 88

The Web of Science Core Collection database was chosen for the quality of the publications and their impact in the development of research regardless of the research field. The analysis showed that most researchers prefer English, but some papers were written in Spanish and French. Still, the number of publications written in these two languages is rather low.

Results: is really helpful knowing which are the most cited sources of funds, authors and journals? Why? The quality of the results is highly affected by the lack of a detailed description of the used methods. I cannot judge the results as long as I do not know how (and why) you selected such articles.

We improved the Results section. We added / modified the following paragraphs:

Lines 94 – 96:

In the last decade, researchers managed to publish up to 201 papers per year, with a mean of 115 papers / year and a lowest of 45 papers in 2010.

Lines 100 – 102:

Due to the important negative effects, floods and their impact on society represents a priority in the work of researchers nowadays. The Web of Sciences database revealed that over 1326 research papers are mainly focused on this topic (Figure 1).

Lines 188 – 191:

The results of the analysis show the interest of researchers to test and validate the methodologies proposed with different research areas (West Africa, Balkans, South and Southeast Asia) [62-64]. Research has also been conducted on a global scale, representing 1.05% [65–67].

Lines 208 – 238:

The role of geospatial technologies - remote sensing and GIS - has been a significant step in flood modeling, forecasting, and hazard assessment. Flood modeling requires the analyst to acquire, maintain, and widely use a spatial database. Remote sensing and GIS are excellent techniques that can meet these requirements.

The role of remote sensing in river basin hydrologic modeling as part of the entire runoff modeling process is due to its ability to provide continuous spatial data, meas-urements of hydrological variables that are not available through traditional techniques but especially its ability to provide global data, e.g., long term.

Modeling and simulation techniques refer to flood modeling as the process of transforming precipitation into a flood hydrograph. In this way, floods are approximated physically or mathematically (using mathematical equations) where the relationships between them represent the state at the system's entry and exit. Geographic Information System (GIS) has increased the importance of remote sensing by improving spatial modeling efficiency. This process has increased the ability to estimate hydrological models.

The hydrological model is a mathematical representation of hydrological processes in a river basin in a simplified form. It was used primarily to understand and explain hydrological processes and for hydrological prediction. This model can be of different types, depending on the kind of approach, deterministic or stochastic. A hydrological model is deterministic if it represents the physical processes in a river basin without considering randomness. It becomes a stochastic model when it incorporates in its random mathematical representation variables and their probability of distribution in the space of parameters.

On the other hand, the hydraulic models used show water flows' evolution to sim-ulate water movement during floods, along waterways, reservoirs, and hydraulic structures.

One of the most used hydrological and hydraulic modeling systems is HEC RAS (River Analysis System). HEC HMS is a generalized modeling system designed to simulate river basin dendritic systems' precipitation-runoff processes with a wide range of applicability.

HEC RAS is an integrated software system designed to operate one-dimensional hydraulic calculations for a complete network of natural and artificial channels.

Lines 252 – 260:

Statistical analysis was used mainly in estimating flood frequency. Until 1966, the most widely used distribution for flood frequency analysis was the type I extreme value distribution, also called the Gumbel distribution, followed by the lognormal. Both of the above distributions are still commonly used, commonly referred to in hydrology as Pearson III type and log-Pearson III type.

Tiago de Oliveira (1982) developed a statistical decision procedure to choose between the Gumbel, Frechet, and Weibull models. The fact is that hydrological data rarely fulfill the classical asymptotic theory's assumptions, and thus the Gumbel, Frechet, or Weibull model is used.

Discussion: many claims reported in this section are not supported by the presented data. Please improve this section, addressing more in detail what you have reported (e.g., the geographical distribution of authors/studies, the funding sources, etc.) and the used methodology.

Thank you for your observations and useful comments. We have added / modified the following paragraphs to the Discussions section.

Lines 297 – 299:

Numerous methodologies that are presented as a novelty are not found in the current practical activity, as there is no validation of them in real-case scenarios.

Lines 305 – 307

New methodologies generally involve the aggregation of complex databases, high res-olution imagery, all generating the need for high-performance IT equipment but also for staff training.

Lines 310 – 312

The paper presents an overview of current research methodologies and identifies the methodological spectrum used in flood analysis, providing researchers with a general overview.

Conclusions: the conclusions are not supported by the rest of the paper. Please, rewrite the whole section.

Thank you for your constructive comments. We improved the Conclusions section by adding the following paragraphs:

Lines 318 – 322:

The present study tried to highlight various aspects, methods and approaches of flood study methodologies between 1979 - 2020.

We analyzed 1326 articles of the most prolific researchers to make a solid and co-herent analysis on the various methods used. The progress in the methodological framework and methodological approaches to floods were examined, highlighting the models used.

Lines 333 – 340:

Advances in geospatial techniques, especially in remote sensing, GIS, 3D hydrological models and machine learning algorithms have revolutionized the methodology of flood analysis.

Multidimensional approach, models based on machine learning, high resolution satellite imagery, hydraulic modelling and selection of efficient flood conditioning pa-rameters are suggested for an in-depth hazard analysis. The use of multidimensional approaches, sophisticated models, site-specific indicators and high-resolution satellite data are areas where further research is required.

Round 2

Reviewer 1 Report

Accept.

Reviewer 2 Report

Dear Authors,

thank you very much for having addressed my concerns.

In my opinion, the present version can be accepted, after a double-checking of the language and rearranging the text following the Journal guidelines.